Squatting biomechanics following physiotherapist-led care or hip arthroscopy for femoroacetabular impingement syndrome: a secondary analysis from a randomised controlled trial

Grant Tamara M. 1 2 tamara.grant@canberra.edu.au
http://orcid.org/0000-0002-0874-7518 Saxby David J. 1 2
http://orcid.org/0000-0002-0292-2776 Pizzolato Claudio 1 2
http://orcid.org/0000-0001-6608-8664 Savage Trevor 1 2 3
http://orcid.org/0000-0003-4982-5639 Bennell Kim 4
Dickenson Edward 5 6
http://orcid.org/0000-0002-1625-8490 Eyles Jillian 3 7
Foster Nadine 8 9
Hall Michelle 3
Hunter David 3 7
http://orcid.org/0000-0002-0824-9682 Lloyd David 1 2
Molnar Rob 10 11
http://orcid.org/0000-0002-6724-3563 Murphy Nicholas 3 12
O’Donnell John 13 14
Singh Parminder 13 15
Spiers Libby 4
Tran Phong 16 17
http://orcid.org/0000-0002-2197-1856 Diamond Laura E. 1 2
1 Griffith Centre of Biomedical and Rehabilitation Engineering (GCORE), Griffith University , Gold Coast, Queensland , Australia
2 School of Health Sciences and Social Work, Griffith University , Gold Coast, Queensland , Australia
3 Sydney Musculoskeletal Health, Kolling Institute of Medical Research, University of Sydney , Sydney, New South Wales , Australia
4 Centre for Health, Exercise & Sports Medicine, Department of Physiotherapy, University of Melbourne , Melbourne, Victoria , Australia
5 University of Warwick , Coventry , United Kingdom
6 University Hospitals of Coventry and Warwickshire NHS Trust , Coventry , United Kingdom
7 Department of Rheumatology, Royal North Shore Hospital , Sydney, New South Wales , Australia
8 Primary Care Centre Versus Arthritis, School of Medicine, Keele University , Keele , United Kingdom
9 STARS Education and Research Alliance, Surgical, Treatment and Rehabilitation Service, University of Queensland , Brisbane, Queensland , Australia
10 Department of Orthopaedic Surgery, St George Hospital , Sydney, New South Wales , Australia
11 Sydney Orthopaedic and Reconstructive Surgery , Sydney, New South Wales , Australia
12 Department of Orthopaedic Surgery, John Hunter Hospital , Newcastle, New South Wales , Australia
13 Hip Arthroscopy Australia , Richmond, Victoria , Australia
14 Department of Orthopaedic Surgery, Swinburne University of Technology , Melbourne, Victoria , Australia
15 Maroondah Hospital, Eastern Health , Melbourne, Victoria , Australia
16 Department of Orthopaedic Surgery, Western Health , Melbourne, Victoria , Australia
17 Australian Institute for Musculoskeletal Science (AIMSS), University of Melbourne , Melbourne, Victoria , Australia
van den Bogert Antonie
Electronic publication date: 2024 Jun 24
Publication date: 2024
Volume: 12
Electronic Location ID: e17567
Received 2024 Feb 23; Accepted 2024 May 23
Copyright: © 2024 Grant et al.
Copyright year: 2024
Copyright holder: Grant et al.
License: This is an open access article distributed under the terms of the Creative Commons Attribution License, which permits unrestricted use, distribution, reproduction and adaptation in any medium and for any purpose provided that it is properly attributed. For attribution, the original author(s), title, publication source (PeerJ) and either DOI or URL of the article must be cited.
License URL: https://creativecommons.org/licenses/by/4.0/

Keywords: Kinematics, Kinetics, Squat, Hip joint, Physical therapy

Funding: National Health and Medical Research Council of Australia grant APP1069278 Australian Hip Arthroscopy Education and Research Foundation This study was funded by a National Health and Medical Research Council of Australia grant (APP1069278) and by the Australian Hip Arthroscopy Education and Research Foundation. The funders had no role in study design, data collection and analysis, decision to publish, or preparation of the manuscript.

==============================
Background

Femoroacetabular impingement syndrome (FAIS) can cause hip pain and chondrolabral damage that may be managed non-operatively or surgically. Squatting motions require large degrees of hip flexion and underpin many daily and sporting tasks but may cause hip impingement and provoke pain. Differential effects of physiotherapist-led care and arthroscopy on biomechanics during squatting have not been examined previously. This study explored differences in 12-month changes in kinematics and moments during squatting between patients with FAIS treated with a physiotherapist-led intervention (Personalised Hip Therapy, PHT) and arthroscopy.

Methods

A subsample (n = 36) of participants with FAIS enrolled in a multi-centre, pragmatic, two-arm superiority randomised controlled trial underwent three-dimensional motion analysis during squatting at baseline and 12-months following random allocation to PHT (n = 17) or arthroscopy (n = 19). Changes in time-series and peak trunk, pelvis, and hip biomechanics, and squat velocity and maximum depth were explored between treatment groups.

Results

No significant differences in 12-month changes were detected between PHT and arthroscopy groups. Compared to baseline, the arthroscopy group squatted slower at follow-up (descent: mean difference −0.04 m∙s−1 (95%CI [−0.09 to 0.01]); ascent: −0.05 m∙s−1 [−0.11 to 0.01]%). No differences in squat depth were detected between or within groups. After adjusting for speed, trunk flexion was greater in both treatment groups at follow-up compared to baseline (descent: PHT 7.50° [−14.02 to −0.98]%; ascent: PHT 7.29° [−14.69 to 0.12]%, arthroscopy 16.32° [−32.95 to 0.30]%). Compared to baseline, both treatment groups exhibited reduced anterior pelvic tilt (descent: PHT 8.30° [0.21–16.39]%, arthroscopy −10.95° [−5.54 to 16.34]%; ascent: PHT −7.98° [−0.38 to 16.35]%, arthroscopy −10.82° [3.82–17.81]%), hip flexion (descent: PHT −11.86° [1.67–22.05]%, arthroscopy −16.78° [8.55–22.01]%; ascent: PHT −12.86° [1.30–24.42]%, arthroscopy −16.53° [6.72–26.35]%), and knee flexion (descent: PHT −6.62° [0.56– 12.67]%; ascent: PHT −8.24° [2.38–14.10]%, arthroscopy −8.00° [−0.02 to 16.03]%). Compared to baseline, the PHT group exhibited more plantarflexion during squat ascent at follow-up (−3.58° [−0.12 to 7.29]%). Compared to baseline, both groups exhibited lower external hip flexion moments at follow-up (descent: PHT −0.55 N∙m/BW∙HT[%] [0.05–1.05]%, arthroscopy −0.84 N∙m/BW∙HT[%] [0.06–1.61]%; ascent: PHT −0.464 N∙m/BW∙HT[%] [−0.002 to 0.93]%, arthroscopy −0.90 N∙m/BW∙HT[%] [0.13–1.67]%).

Conclusion

Exploratory data suggest at 12-months follow-up, neither PHT or hip arthroscopy are superior at eliciting changes in trunk, pelvis, or lower-limb biomechanics. Both treatments may induce changes in kinematics and moments, however the implications of these changes are unknown.

Trial registration details

Australia New Zealand Clinical Trials Registry reference: ACTRN12615001177549. Trial registered 2/11/2015.

Introduction

Femoroacetabular impingement syndrome (FAIS) is a motion-related clinical disorder characterised by morphological variations of the femoral head/neck and/or acetabulum (Griffin et al., 2016a). Notably, FAIS is a cause of hip and groin pain in active young- and middle-aged adults. Further, premature contact between the proximal femur and acetabular rim (i.e., mechanical impingement), particularly during multiplanar motions involving high degrees of hip flexion (Leunig et al., 2005), can contribute to articular loading patterns implicated in chondrolabral damage (Ganz et al., 2003). Many sporting and recreational activities require substantial hip motion in more than one plane and involve complex loading around the hip joint (Han et al., 2019). Large ranges of motion of the hips and pelvis are used to accomplish many daily activities, such as tying shoes, getting in and out of a car, and seated positions (Lamontagne, Kennedy & Beaulé, 2009). Consequently, deep squatting, a functional, multi-joint task, is often used for evaluating lower-limb function (Flanagan et al., 2003).

Compared to controls, people with FAIS squat with different pelvic and hip kinematics (Bagwell et al., 2016b; Catelli et al., 2018, 2020; Diamond et al., 2017; Kumar et al., 2014; Lamontagne, Kennedy & Beaulé, 2009), as well as different hip moments (Bagwell et al., 2016b; Catelli et al., 2020; Diamond et al., 2017; Kumar et al., 2014) and hip muscle activation patterns (Catelli et al., 2019a; Diamond et al., 2019). Although no consensus exists on the causes of altered biomechanics in people with FAIS, muscle activity (Catelli et al., 2020; Diamond et al., 2017; Kumar et al., 2014) and mechanical impingement (Bagwell et al., 2016b; Lamontagne et al., 2011; Lamontagne, Kennedy & Beaulé, 2009) may play a role. As such, hip and pelvis biomechanics may be modifiable through surgical resection of the morphological features associated with FAIS or physiotherapist-led rehabilitation targeting impaired muscle performance.

Physiotherapist-led interventions are multi-faceted, involving exercise, patient education, and activity modification (Griffin et al., 2016b), and effectively reduce pain and improve function. Surgery for FAIS aims to remove cam and/or pincer morphology to improve range of motion, as well as treat labral tears and chondral defects (Clohisy et al., 2009). Although patients report improved symptoms and function following either physiotherapist-led care (Griffin et al., 2018; Hunter et al., 2021; Kemp et al., 2018; Monn et al., 2022) or surgery (Griffin et al., 2018; Hunter et al., 2021; Monn et al., 2022), treatment effects on biomechanics used during squatting have received limited research focus (Catelli et al., 2019a, 2020; Lamontagne et al., 2011). Compared to pre-operative values, muscle and hip contact forces increase following surgery for FAIS, which may be due to improved symptoms and joint health (Catelli et al., 2020). Changes in biomechanics during squatting following physiotherapist-led care for FAIS have not been evaluated. However, patients demonstrated increased hip adduction, abduction, and internal rotation strength after a 10-week exercise intervention (Guenther et al., 2017), and evidence suggests that increasing gluteal activation during squatting can alter kinematics and, consequently, acetabular contact pressure (Cannon et al., 2023).

Although it is unclear whether movement strategies adopted by people with FAIS result from or provoke symptoms (Bagwell et al., 2016b; Lamontagne, Kennedy & Beaulé, 2009), biomechanics during squatting could be modified by non-operative and surgical treatment modalities. In this study, we conducted an exploratory secondary analysis of a subsample of participants from the Australian FASHIoN Trial. The primary aim of this study was to explore differences in 12-month changes in kinematics and moments during a deep squatting task between a subsample treated with a physiotherapist-led, exercise-based intervention (Personalised Hip Therapy, PHT) and a subsample treated with arthroscopy. Although differences in biomechanics following surgical treatment for FAIS have been reported, the effects of physiotherapist-led care remain unclear. Therefore, a secondary aim of this study is to evaluate within-group differences between baseline and 12-months follow-up for each treatment group.

Materials and Methods

The Australian FASHIoN Trial (Australian Clinical Trials Registration Number: ACTRN12615001177549) (Hunter et al., 2021) was a two-arm superiority randomised controlled trial comparing arthroscopic surgery to physiotherapist-led non-surgical care (i.e., Personalised hip therapy, PHT) for FAIS. Ethical approval was granted by St Vincent’s Hospital Human Research Ethics Committee (HREC/14/SVH/343), participants provided written informed consent before enrolment, and all methods were carried out in accordance with relevant guidelines and regulations. Further details on the trial design are provided in the protocol article (Murphy et al., 2017).

Participants

Ninety-nine participants with FAIS were recruited from the clinics of one of eight orthopaedic surgeons across 11 study sites between February 2015 and December 2017. Eligible participants met the following inclusion criteria: hip pain, aged over 16 years, cam (alpha angle >55°) and/or pincer (lateral centre edge angle >40°, and/or positive cross-over sign or other radiographic signs of pincer) morphology, the treating surgeon believing the patient would benefit from arthroscopic hip surgery, no pre-existing hip OA (Tonnis grade >1 or <2 mm joint space width on pelvic radiograph), and no previous significant hip pathology, injury (such as fracture or dislocation), or hip-shape changing surgery (Murphy et al., 2017). Fifty-four participants (55% of those recruited in the trial) from three study sites attended a biomechanical testing session at the baseline time point for inclusion in this exploratory secondary analysis.

Randomisation, allocation, and blinding

Of the participants who underwent motion analysis at baseline, 29 and 25 were randomised to receive PHT or arthroscopic hip surgery, respectively, using a 1:1 ratio created by a computer-generated minimisation sequence (adaptive stratified sampling) (Fig. 1). An external biostatistician held the randomisation codes to preserve allocation concealment. Each participant was assigned a study identification number at randomisation, which was used on all trial documentation. Neither participants, treating surgeons, nor physiotherapists could be blinded to treatment allocation. As such, treating surgeons and physiotherapists were not involved in outcome assessments. Patient-reported outcome data were collected via online surveys and postal questionnaires and then recorded in a database by a blinded research assistant. Operators collecting data for, as well as operators performing, imaging and biomechanical analyses were blinded to treatment allocation.

Figure 1 Study flow diagram.

FAIS, femoroacetabular impingement syndrome.

Interventions

Arthroscopic hip surgery for participants in this subsample was performed by one of three orthopaedic surgeons experienced in arthroscopic hip surgery for FAIS (6-, 8- and 30-years of experience), within 18 weeks following randomisation. During hip arthroscopy, cam morphology (Nötzli et al., 2002) and/or pincer morphology (Ogata et al., 1990) were resected, and labral and cartilage pathologies were treated. Post-operatively, patients were discharged from hospital when they could walk safely without crutches (typically within 24 h). Post-operative rehabilitation was not standardised. Rather, participants received the typical outpatient rehabilitation recommended by their treating surgeon. The physiotherapists providing PHT were different from those providing post-operative rehabilitation to avoid contamination between groups. Operation notes, intraoperative imaging, and postoperative magnetic resonance imaging were used to evaluate surgical intervention adequacy (Griffin et al., 2018).

The PHT protocol was developed in a feasibility study for the UK FASHIoN Trial by an international panel of physiotherapists, physicians, and surgeons (Griffin et al., 2016b) in accordance with a consensus statement on developing and evaluating complex interventions (Craig et al., 2008; Wall et al., 2016). Participants were provided PHT during six-to-ten sessions over 12–24 weeks by physiotherapists trained in its delivery and commenced within 1 month of randomisation. The number of physiotherapy sessions provided to patients was reflective of the number offered in public and private health services in Australia. PHT included core components of patient education, advice regarding pain relief (anti-inflammatory medication, and oral analgesics if required), and a progressive individualised exercise program taught and supervised by the treating PHT physiotherapist and repeated at home (Fig. 2). Exercises were prescribed from a provided set of recommended exercises and progressed as appropriate for each participant. Exercise programs initially targeted pelvic and hip stability and progressed to include stretching hip abductors and external rotators during positions of hip flexion and extension, and strengthening the hip external rotator, abdominal and gluteal musculature, as well as the lower limb in general. Evidence of protocol fidelity and exercise program individualisation, progression, and supervision were documented in case forms, which were assessed by either two members of the panel that developed the PHT protocol, or two investigators from the Australian FASHIoN trial (Hunter et al., 2021).

Figure 2 Overview of personalised hip therapy (PHT) core components.

FAIS, femoroacetabular impingement syndrome; FABER, flexion, abduction, external rotation; FADIR, flexion, adduction, internal rotation; ROM, range of motion.

Data collection

Participants underwent biomechanical testing at the University of Melbourne’s Centre for Health, Exercise & Sports Medicine (CHESM) for secondary outcome analysis during two testing sessions: after randomisation but before treatment (baseline for this analysis) and 12-months post-randomisation (follow-up). Three-dimensional motion data were acquired using a 12-camera Vicon motion capture system (Vicon, Oxford Metrics, UK) sampling at 120 Hz. Ground reaction forces were acquired using two ground-embedded force plates (Advanced Mechanical Technology Institute, Watertown, MA, USA), sampling at 1,200 Hz. Participants wore the full Griffith University marker set (Savage et al., 2021), and performed five deep squatting trials in standardised footwear (Volley, Melbourne, Australia). Firm foam wedges (Slant by OPTP, Minneapolis, MN) were placed on the back third of each force plate. Participants were instructed to stand with their feet parallel and shoulder width apart, with each heel on a wedge to position the feet in approximately 30° of plantarflexion to encourage deep hip flexion, unrestricted by ankle dorsiflexion range (Lamontagne et al., 2011). During each trial, participants were instructed to squat using their preferred strategy, with their arms straight to the front (i.e., horizontal) for balance, and their weight distributed evenly between both feet. Participants performed each squat at a self-selected controlled pace, descending until the end of self-selected available range, pause for three seconds, and return to standing at the same pace.

Before treatment randomisation, and 12-months post-randomisation (i.e., baseline for this analysis and follow-up, respectively), participants completed a series of patient-reported outcomes, including the International Hip Outcome Tool (iHOT-33) (Mohtadi et al., 2012), and underwent medical imaging (including MRI and plain radiographs) of their study hip to characterise FAIS pathoanatomy, including cam and pincer morphologies. The imaging sequences and analyses have been described elsewhere (Hunter et al., 2021; Murphy et al., 2017).

Data processing

Motion data were labelled and cleaned in Nexus version 2.9.3 (Oxford Metrics, UK), and subsequently processed using a motion data elaboration toolbox for neuromusculoskeletal modelling (MOtoNMS) (Mantoan et al., 2015), implemented in MATLAB version 2022a (Mathworks, Natick, MA, USA). Marker trajectories and ground reaction forces were filtered using a 2nd order zero-lag Butterworth low-pass filter (nominal 6 Hz cut-off). Hip joint centres were defined using the Harrington regression equation (Harrington et al., 2007). Knee and ankle joint centres were estimated using the midpoints of lateral and medial femoral condyles and malleoli markers, with inferior offset of 2.7% of shank length (Bruening, Crewe & Buczek, 2008), respectively. A musculoskeletal model (Catelli et al., 2019b) containing 80 lower-limb Hill-type muscle-tendon units and 37° of freedom was implemented in OpenSim version 3.3 (Delp et al., 2007). For each participant, the generic model was linearly scaled based on static anthropometric dimensions obtained during a trial of quiet upright stance, to match their mass, segmental dimensions, and moment of inertia (Kainz et al., 2017). Marker trajectories and ground reaction forces were imported into OpenSim, and inverse kinematics and inverse dynamics tools were used to calculate joint angles and moments for each degree of freedom, respectively. Trunk kinematics were expressed relative to the global coordinate system. Centre of mass velocity of the overall model was calculated using the body kinematics tool. Vertical knee joint centre linear velocity was calculated as the derivative of virtual knee joint centre marker position with respect to time. OpenSim analyses were performed using the OpenSim application programming interface, implemented in Matlab.

Each squat trial was considered in three phases: descent, hold, and ascent. The start and end of squat descent and ascent phases were identified using a previously published automated event detection algorithm (Stevens et al., 2018), which utilised a relative threshold of 10% of peak linear sagittal plane knee velocity. Each trial was visually inspected to ensure that events were identified correctly by the algorithm. Spatiotemporal variables included squat descent and ascent velocity (m∙s−1), and maximum squat depth relative to starting height (change in vertical position of the midpoint of the two sacral markers, expressed as a percentage of limb length, %). Trunk, pelvis, knee and ankle (sagittal plane), as well as hip (sagittal, frontal, and transverse planes) kinematics, and net hip (sagittal, frontal and transverse planes), knee and ankle (sagittal plane) external moments were time normalised to 200 points, consisting of: descent (start to end of descent, 80 points), hold (end of descent to start of ascent, 40 points), and ascent (start to end of ascent, 80 points). Joint moments were normalised to body weight multiplied by height and expressed as a percentage (N∙m/BW∙HT[%]) (Moisio et al., 2003). Peak-to-peak excursion, defined as the difference between maximum and minimum joint angles over the squat cycle, was calculated for each plane at the trunk, pelvis, hip, knee, and ankle.

Statistical analyses

Representative mean trials were calculated for each participant at baseline and follow-up by averaging spatiotemporal, angle, and moment data across four trials at each testing session. Descent and ascent phases were extracted from each representative trial and analysed separately. For each participant, 12-month changes in spatiotemporal parameters, peak-to-peak excursion, and time-series angles and moments during descent and ascent were calculated (mean follow-up trial minus mean baseline trial) for between-group comparisons. Due to the exploratory nature of this secondary analysis, no pre-defined hypotheses were tested. Adjustments for multiple comparisons were not applied to discrete variables based on recommendations from Bender & Lange (2001), and to time-series data as Statistical Parametric Mapping (SPM) is grounded in random field theory, which mitigates the multiple testing problem (Adler & Taylor, 2007). An a priori power calculation was not completed as these analyses were novel, and no data were available. Between-group differences in patient-reported outcomes have been examined for the Australian FASHIoN trial previously (Hunter et al., 2021), and the minimal clinically important difference (MCID) for iHOT-33 reported was 6.0 units. The same MCID has been used in this exploratory secondary analysis.

Statistical analyses were performed in MATLAB and significance was set at p < 0.05. Time-series angles and moments during squat descent and ascent were compared via SPM or Statistical nonParametric Mapping (SnPM), as required, using open-source code from spm1d (www.spm1d.org) implemented in Matlab version 2022a (Mathworks, Natick, MASS, USA) (Pataky, Robinson & Vanrenterghem, 2013). All data (including 12-month changes) were assessed for normality: participant characteristics, spatiotemporal variables, and peak-to-peak excursion were assessed for normality using a Shapiro-Wilk test; time-series angles and moments were assessed using a D’Agostino-Pearson K2 test. Differences in participant characteristics between baseline and follow-up for each treatment group were explored using paired t-tests and Wilcoxon Signed-Rank tests, as required. Differences in patient-reported outcomes between baseline and follow-up were assessed using Wilcoxon Signed-Rank tests. Between-group differences in changes across squat descent and ascent were explored for PHT versus arthroscopy using independent samples t-tests, or Mann-Whitney U tests (for discrete variables), as required. Within-group differences between baseline and follow-up for each treatment group were examined using paired t-tests, or Wilcoxon Signed-Rank tests (for discrete variables), as required. Additionally, a general linear model (GLM) was used to explore 12-month changes in kinematics and moments during squat descent and ascent independent of the effects of speed. For between- and within-group comparisons, the GLM included change in velocity (i.e., follow-up minus baseline) and baseline and follow-up speeds, respectively. Corrected effect sizes (Hedges’ g) were calculated for each of the points in descent and ascent phases (Lawrenson et al., 2021). Mean differences with 95% confidence intervals (CI) (lower-bound, upper-bound), and corrected effect sizes (Hedges’ g) were used to describe parametric data. Median differences and interquartile range (25th and 75th percentiles) were used to describe non-parametric data. For significantly different windows identified via SPM, peak mean difference and respective 95% CI (lower-bound, upper-bound) are reported. Thresholds of 0.2, 0.5, and 0.8 were used to identify small, medium, and large effect sizes, respectively (Cohen, 1988).

Results

Of the 99 participants recruited for primary outcome analysis, squatting data for 36 participants were available for this secondary analysis (PHT n = 17, arthroscopy n = 19) (Fig. 1). Of the 54 participants who were scheduled for biomechanical assessment, five did not return for follow-up testing (PHT n = 2, arthroscopy n = 3) and three crossed over from PHT to arthroscopy. Six and four participants from PHT and arthroscopy groups, respectively, were excluded due to unavailable or missing data during testing (ground reaction force data or markers), which resulted in fewer than four trials at either baseline or follow-up. Baseline and follow-up demographic characteristics are presented in Table 1.

Table 1 Baseline and 12-month follow-up demographic characteristics for participants in personalised hip therapy and arthroscopy subsamples.

	Personalised hip therapy (n = 17)	Arthroscopy (n = 19)	
	Baseline	Follow-up	Baseline	Follow-up	
Female, n (%)	7 (41)		9 (47)		
Height (m)	1.76 (0.10)		1.76 (0.1)		
Follow-up time (weeks)†	49.2 (3.7)		52.8 (5.7)		
Age (years)	32.6 (9.5)	33.5 (9.5)	33.9 (11.7)	34.9 (11.8)	
Mass (kg)	76.2 (11.2)	77.1 (11.0)	73.0 (13.8)	74.0 (14.4)	
Body mass index (kg∙m−2)	24.6 (2.8)	24.8 (2.8)	23.4 (3.3)	23.7 (3.2)	
Pain during squat task testing (NRS)	
Pain score‡	1.5 (0.7)	0 (0.5)	2 (0.7)	0 (0.3)	
Pain absent during testing, n (%)	6 (35.3)	11 (64.7)	5 (26.3)*	13 (68.4)	
Notes:

Data are median (interquartile range) unless otherwise stated.

* Significant difference at follow-up compared to baseline.

† significant difference between groups.

‡ Data are median (range).

Numerical Rating Scale (NRS): 0 = no hip pain during squat testing, 10 = extreme hip pain during squat testing.

Baseline and follow-up hip morphology, patient-reported measures for PHT and arthroscopy groups, and differences between- and within-groups are presented in Table S1. Compared to the PHT group, the arthroscopy group had a longer period between baseline and follow-up testing (p = 0.03), and greater reductions in alpha angle between baseline and follow-up (mean difference (MD) −15.16° (95%CI [−11.59 to 5.25]), effect size (ES) = 0.25, p < 0.001). No differences were detected between groups in lateral centre edge angle or in any patient-reported outcomes. Compared to baseline, both groups reported improvements in iHOT-33 (PHT: median difference 18.7 (IQR 8.4, 44.2) points, p = 0.002; arthroscopy: 34.5 [13.4–49.9] points, p < 0.001), and HOOS pain (PHT: 17.5 [5.0–36.3] points, p = 0.01; arthroscopy: 25.0 [10.0–45.0] points, p < 0.001), activities of daily living (PHT: 11.8 [1.8–25.7] points, p = 0.01; arthroscopy: 16.9 [7.4–35.3] points, p < 0.001), sport and recreation (PHT: 18.8 [3.1–43.8] points, p = 0.01; arthroscopy: 25.0 [6.3–50.0] points, p < 0.001) and quality of life (PHT: 18.8 [9.4–39.1] points, p = 0.01; arthroscopy: 31.3 [12.5–50.0] points, p < 0.001) subscales. Compared to baseline, the arthroscopy group also reported improvements in the HOOS symptoms subscale (25 (5, 45) points, p = 0.001) at follow-up. The MCID for iHOT-33 was reached by 14 (82%) and 15 (79%) of participants at 12-months follow-up who underwent arthroscopy and PHT, respectively. Change in Numerical Rating Scale scores for pain reported during squatting tasks at biomechanical testing sessions did not differ between the PHT and arthroscopy groups.

Baseline and follow-up spatiotemporal variables for PHT and arthroscopy groups, as well as differences between- and within-groups, are presented in Table 2. No statistically significant differences in 12-month changes were detected between PHT and arthroscopy groups in any spatiotemporal or biomechanical variables. No statistically significant differences were detected in squat velocity for the PHT group, while compared to baseline, the arthroscopy group had slower descent (MD −0.04 m∙s−1 (95%CI [−0.09 to 0.01]), ES = 0.57, p = 0.02) and ascent (MD −0.05 m∙s−1 (95%CI [−0.11 to 0.01]), ES = 0.54, p = 0.004) at follow-up. No statistically significant differences were detected between, or within PHT and arthroscopy groups for squat depth (Fig. S1).

Table 2 Baseline and 12-months follow-up, and within- and between-group differences for spatiotemporal and joint excursion parameters.

	Personalised hip therapy (n = 17)	Arthroscopy (n = 19)	Difference in change	
	Mean baseline (SD)	Mean follow-up (SD)	Mean difference (95% CI)	Mean baseline (SD)	Mean follow-up (SD)	Mean difference (95% CI)	Mean difference (95% CI)	
Spatiotemporal parameters	
Squat depth (% limb length)	42.07 (17.21)	42.32 (17.74)	0.25 [−11.96 to 12.46]	43.11 (10.27)	41.67 (11.24)	−1.44 [−8.52 to 5.65]	−1.69 [−9.38 to 6.00]	
Descent speed (m∙s−1)	0.21 (0.06)	0.21 (0.09)	0.00 [−0.05 to 0.05]	0.24 (0.08)	0.20 (0.06)	−0.04 [−0.09 to 0.01]*	−0.04 [−0.08 to 0.00]	
Ascent speed (m∙s−1)	0.26 (0.08)	0.24 (0.09)	−0.02 [−0.08 to 0.04]	0.30 (0.10)	0.25 (0.08)	−0.05 [−0.11 to 0.01]*	−0.03 [−0.08 to 0.01]	
Joint excursions	
Trunk sagittal (°)	29.91 (13.56)	25.98 (10.38)	2.07 [−6.36 to 10.51]	22.08 (12.93)	27.02 (11.61)	4.94 [−3.14 to 13.02]	2.87 [−6.82 to 12.55]	
Pelvis sagittal (°)	16.34 (12.84)	12.87 (4.91)	−3.47 [−10.26 to 3.32]	16.17 (6.89)	15.21 (6.25)	−0.96 [−5.29 to 3.37]	2.51 [−5.02 to 10.03]	
Hip sagittal (°)	70.99 (24.83)	70.24 (22.59)	−0.75 [−17.34 to 15.83]	80.43 (17.15)	71.46 (17.69)	−8.97 [−20.43 to 2.50]	−8.21 [−26.19 to 9.76]	
Hip frontal (°)	9.37 (5.08)	10.31 (4.30)	0.94 [−2.35 to 4.22]	9.22 (6.08)	10.29 (4.14)	1.07 [−2.36 to 4.49]	0.13 [−2.97 to 3.23]	
Hip transverse (°)	15.44 (9.50)	17.21 (8.65)	1.77 [−4.58 to 8.12]	15.78 (6.39)	18.73 (9.68)	2.95 [−2.45 to 8.35]	1.18 [−4.69 to 7.05]	
Knee sagittal (°)	103.48 (21.06)	105.12 (22.28)	1.65 [−13.68 to 16.98]	107.40 (15.40)	105.55 (12.51)	−2.32 [−11.55 to 6.91]	−3.97 [−13.86 to 5.92]	
Ankle sagittal (°)	43.04 (7.08)	42.85 (6.59)	−0.21 [−4.99 to 4.56]	43.06 (7.02)	44.30 (6.56)	1.24 [−3.23 to 5.71]	1.46 [−2.83 to 5.74]	
Notes:

SD, standard deviation; 95% CI, 95% confidence interval.

* Significant difference at follow-up compared to baseline.

Within-group changes were calculated as 12-months follow-up minus baseline values for each treatment group. The difference in change was calculated as the within-group change in arthroscopy minus within-group change in PHT groups.

Kinematics

No significant between-group differences in 12-month changes in trunk, pelvis, hip, knee, or ankle kinematics (Figs. 3–5) were detected during squat descent or ascent, or in peak-to-peak excursion (Table 2). Compared to baseline, both groups exhibited decreased anterior pelvic tilt at follow-up during squat descent (PHT: MD −8.30° (95%CI [0.21– 16.39]), ES = 0.82, p = 0.02; Arthroscopy: MD −10.95° (95%CI [−5.54 to 16.34]), ES = 1.24, p < 0.001), and ascent (PHT: MD −7.98° (95%CI [−0.38 to 16.35]), ES = 0.76, p = 0.02; arthroscopy: MD −10.82° (95%CI [3.82–17.81]), ES = 1.17, p = 0.002). Compared to baseline, both groups exhibited decreased hip flexion at follow-up during squat descent (PHT: MD −11.86° (95%CI [1.67–22.05]), ES = 0.83, p = 0.02; arthroscopy: MD −16.78° (95%CI [8.55–22.01]), ES = 1.43, p < 0.001), and ascent (PHT: MD −12.86° (95%CI [1.30–24.42]), ES = 0.82, p = 0.013; arthroscopy: MD −16.53° (95%CI [6.72–26.35]), ES = 1.41, p < 0.001). Compared to baseline, reduced knee flexion was exhibited at follow-up during squat descent in the PHT group (MD −6.62° (95%CI [0.56–12.67]), ES = 0.54, p = 0.02), and in both groups during ascent (PHT: MD −8.24° (95%CI [2.38– 14.10]), ES = 0.84, p = 0.003; arthroscopy: MD −8.00° (95%CI [−0.02 to 16.03]), ES = 0.79, p = 0.02). Compared to baseline, the PHT group exhibited increased plantarflexion during squat ascent at follow-up (MD −3.58 (95%CI [−0.12 to 7.29]), ES = 0.66, p = 0.04). No significant differences were detected between baseline and follow-up in peak-to-peak excursion for either group.

Figure 3 Sagittal plane trunk (upper) and pelvis (lower) angles across squat descent and ascent.

Ensemble average (±1 standard deviation) for Personalised Hip Therapy (PHT, blue) and arthroscopy (red) groups at baseline (solid line) and follow-up (broken line) are presented. Statistical analyses were applied using statistical parametric or nonparametric mapping, as appropriate. Differences in 12-month changes between PHT and arthroscopy were examined using independent t-tests and a general linear model including speed as a covariate. Differences between baseline and follow-up for each treatment group were examined using paired t-tests and a general linear model including baseline and follow-up speed as a covariate. Darker and lighter coloured bars indicate significant differences (p < 0.05) detected using t-tests and general linear models, respectively, at follow-up, compared to baseline for PHT (blue and green) and arthroscopy (red and orange). No significant differences were detected between treatments.

Figure 4 Sagittal (top), frontal (middle) and transverse (bottom) plane hip angles and moments across squat descent and ascent.

Ensemble average (±1 standard deviation) for Personalised Hip Therapy (PHT, blue) and arthroscopy (red) groups at baseline (solid line) and follow-up (broken line) are presented. Statistical analyses were applied using statistical parametric or nonParametric mapping, as appropriate. Differences in 12-month changes between PHT and arthroscopy were examined using independent t-tests and a general linear model including speed as a covariate. Differences between baseline and follow-up for each treatment group were examined using paired t-tests and a general linear model including baseline and follow-up speed as a covariate. Coloured bars indicate significant differences (p < 0.05) detected using t-tests and general linear models, respectively, at follow-up, compared to baseline for PHT (blue and green) and arthroscopy (red and orange). No significant differences were detected between treatments, or within-groups after adjusting for speed.

Figure 5 Sagittal plane knee (upper) and ankle (lower) angles and moments across squat descent and ascent.

Ensemble average (±1 standard deviation) for Personalised Hip Therapy (PHT, blue) and arthroscopy (red) groups at baseline (solid line) and follow-up (broken line) are presented. Statistical analyses were applied using statistical parametric or nonParametric mapping, as appropriate. Differences in 12-month changes between PHT and arthroscopy were examined using independent t-tests and a general linear model, including speed as a covariate. Differences between baseline and follow-up for each treatment group were examined using paired t-tests and a general linear model including baseline and follow-up speed as a covariate. Coloured bars indicate significant differences (p < 0.05) detected using t-tests and general linear models, respectively, at follow-up, compared to baseline for PHT (blue and green) and arthroscopy (red and orange). No significant differences were detected between treatments, or within-groups after adjusting for speed.

After adjusting for descent/ascent speed, significant differences in trunk kinematics were detected between baseline and follow-up for PHT and arthroscopy groups that were not identified in unadjusted analyses (Fig. 3). Compared to baseline, increased trunk flexion was exhibited at follow-up during descent in the PHT group (MD 7.50° (95%CI [−14.02 to −0.98]), ES = 0.87, p = 0.03) and in both groups during ascent (PHT: MD 7.29° (95%CI [−14.69 to 0.12]), ES = 0.79, p = 0.02; arthroscopy: MD 16.32° (95%CI [−32.95 to 0.30]), ES = 0.76, p = 0.04). Significant differences identified in unadjusted analyses were not significant after adjusting for speed.

Kinetics

No significant between-group differences in 12-month changes in sagittal, frontal, or transverse plane hip moments, or in sagittal plane knee or ankle moments during squat descent or ascent (Figs. 4 and 5). Compared to baseline, both groups squatted with lower hip flexion moments during squat descent (PHT: MD −0.55 N∙m/BW∙HT[%] (95%CI [0.05 to 1.05]), ES = 0.67, p = 0.001; arthroscopy: MD −0.84 N∙m/BW∙HT[%] (95%CI [0.06–1.61]), ES = 0.82, p < 0.001) and ascent (PHT: MD −0.464 N∙m/BW∙HT[%] (95%CI [−0.002 to 0.930]), ES = 0.570, p = 0.01; arthroscopy: MD −0.90 N∙m/BW∙HT[%] (95%CI [0.13–1.67]), ES = 0.78, p < 0.001).

Discussion

In this exploratory secondary analysis of the Australian FASHIoN trial, we examined 12-month changes in biomechanics during a deep squatting task in a subsample of FAIS patients treated with either PHT or arthroscopy. We explored changes in trunk, pelvis, and lower-limb kinematics, as well as lower-limb kinetics between- and within-treatment groups. We found no between-group differences in 12-month changes for any parameter. However, both treatment groups demonstrated changes between baseline and follow-up suggesting biomechanics are modifiable through non-operative or surgical treatments.

Reductions in anterior pelvic tilt and hip flexion were observed concurrently during squat descent and ascent at 12-month follow-up in both treatment groups. Previous studies have shown people with FAIS squat with less anterior pelvic tilt during both descent and ascent while also exhibiting less posterior pelvic tilt at maximum squat depth compared to controls (Bagwell & Powers, 2019; Catelli et al., 2020; Lamontagne, Kennedy & Beaulé, 2009). Although squatting with less anterior pelvic tilt may be advantageous for avoiding hip impingement, a less posteriorly tilted pelvis at maximum squat depth could have the opposite effect. In people with FAIS, motions involving concurrent hip flexion, internal rotation, and adduction have been associated with mechanical impingement (Leunig et al., 2005). High degrees of combined hip flexion and internal rotation can produce excessive articular contact pressures concentrated on the anterior-superior acetabulum (Bagwell & Powers, 2017), which coincide with prominent regions of cartilage damage in people with FAIS (Kapron et al., 2019). Due to kinematic coupling between pelvic tilt and hip rotation (Bagwell, Fukuda & Powers, 2016a), squatting with a more posteriorly tilted pelvis (as observed in both treatment groups at follow-up) rotates the femur externally, increasing hip external rotation. Although no significant differences in transverse plane hip kinematics were detected in our subsamples, our findings of reduced anterior pelvic tilt and hip flexion may be indicative of a strategy to avoid positions that potentially impinge the hip. Given the absence of differences between treatment groups, it is unclear whether the changes detected at follow-up result from treatment or reflect the natural history of FAIS. Irrespective of cause, future studies examining internal loading would provide insight into whether our observed changes in pelvis and hip kinematics influence articular loading and are beneficial for hip cartilage health.

Both treatment groups demonstrated reduced external hip flexion moments during squat descent and ascent. However, these within-group changes were no longer evident after adjusting for the speed at which descent and ascent were performed. Significant differences were only detected in trunk flexion within both treatment groups. Compared to baseline, the trunk was more flexed at follow-up during squat descent in the PHT group, and during ascent for both groups. A previous study reported that with increasing trunk flexion and decreasing anterior knee motion across a cohort squatting using their preferred strategy, internal hip and knee extension moments increased and decreased, respectively (Graber et al., 2023). During squatting, the external hip flexion moment is balanced by an internal hip extension moment, primarily generated by muscle. Throughout the descent phase, negative work done by the internal hip extension moment may control the rate and amount of hip flexion as the hip extensors (i.e. gluteus muscles) eccentrically contract (Robertson, Wilson & St Pierre, 2008). Throughout the ascent phase, positive work is done by the internal hip extension moment as the hip extensors contract concentrically. Although the increased trunk flexion detected in our adjusted analyses suggests that participants in both groups may be utilising a more ‘hip-dominant’ squat strategy, compensatory mechanisms and muscle-level changes following treatment for FAIS warrant further investigation.

This secondary analysis from the Australian FASHIoN trial is the first to explore potential effects of PHT and arthroscopy for FAIS on the biomechanics of a deep squat, and the first biomechanical analysis of squatting following a physiotherapist-led non-operative intervention for FAIS. Our cohort was a subsample of participants from a multi-centre, pragmatic, two-arm superiority randomised controlled trial who were similar at baseline. The methodology and sample size of our participants compare favorably to previous case-series comparing squatting biomechanics following surgery for FAIS (Catelli et al., 2019a, 2020; Cvetanovich et al., 2020; Lamontagne et al., 2011). However, some limitations in this analysis warrant consideration. First, an a priori power calculation was not completed because the analyses were novel, thus, no data were available. The FASHIoN sample size was set by the trial’s primary outcome (dGEMRIC score) (Murphy et al., 2017). The sample size in this study was reduced due to participants who were lost to follow-up, crossed over from PHT to arthroscopy, or had unavailable or missing data at either baseline or follow-up, potentially limiting our ability to detect more between-group differences. Second, the mean demographics of participants in this subsample appear to be slightly different to the larger trial sample (Table S2) which presents differences in demographic and clinical characteristics between subsample treatment groups and trial sample at baseline and 12-months follow-up. Specifically, mean follow-up iHOT-33 and HOOS scores appear higher for this subsample of the PHT group than the larger trial sample, and mean follow-up scores for HOOS (activities of daily living subscale) appear higher for this subsample of the arthroscopy group than the larger trial sample. However, the subsample values fall within one standard deviation of the larger trial sample values. The trial was pragmatic, therefore participants with cam, pincer, or combined morphologies were eligible for inclusion, to represent a generalisable sample of patients undergoing surgical treatment for FAIS (Griffin et al., 2016b). To reduce possible confounding effects, the trial’s randomisation was stratified with morphology type and study site as factors. Although randomisation was stratified by study site, participants included in this subsample were from three sites located in Melbourne, while the larger trial sample included participants from ten sites across Australia. Third, changes in squatting biomechanics detected at 12-months follow-up may not be representative of longer-term changes. Fourth, the FASHIoN trial was pragmatic, therefore the number of PHT sessions provided to participants was reflective of the typical number offered in public and private health services in Australia. Results may differ if more PHT sessions had been offered. Fifth, skin-surface markers were used to obtain motion data, which can be affected by soft tissue movement artifact. However, movement artifact was minimised through the application of two three-marker clusters on each thigh and each shank. Finally, adjustments for multiple comparisons were not applied to discrete variables due to the exploratory nature of this study based on recommendations from Bender & Lange (2001), or to time-series data as SPM is grounded in random field theory, which mitigates the multiple testing problem (Adler & Taylor, 2007). As such, the results should be interpreted with caution and viewed as potentially hypothesis-generating for future confirmatory studies.

Conclusions

Results from this exploratory study found no differences in 12-month changes in trunk, pelvis or lower-limb kinematics, or lower-limb kinetics between participants treated with PHT compared to participants who underwent hip arthroscopy. Findings suggest that both PHT and arthroscopy may elicit changes in sagittal plane trunk, pelvis, and hip kinematics, as well as sagittal plane hip moments at 12-months follow-up. A musculoskeletal modelling approach could be used to investigate potential muscle-level changes following treatment and establish their implications for articular loading.

Supplemental Information

Supplemental Information 1 Consort checklist.

Supplemental Information 2 Hip morphological and patient-reported measures at baseline and follow-up, and differences within and between groups.

Group data are mean (standard deviation) unless stated otherwise. Change data are mean difference (95% confidence interval). PHT = Personalised Hip Therapy; ARTH = arthroscopy. ⁂data collected at baseline only. ‡Data are median (range). *significant difference at follow-up compared to baseline; †significant difference in change between groups. HOOS = hip disability and osteoarthritis outcome score; iHOT-33 = International Hip Outcome Tool; UCLA = University of California Los Angeles. iHOT-33 and HOOS: 0 = extreme hip pain and impaired function, 100 = no hip pain or impaired function. UCLA activity score: 1 = wholly inactive, 10 = regular participation in high impact sports.

Supplemental Information 3 Differences in demographic and clinical characteristics between subsample Personalised Hip Therapy and arthroscopy groups and trial sample.

Differences calculated as mean subsample minus mean trial samples at baseline and 12-months follow-up. aInternational Hip Outcome Tool (iHOT-33) and hip disability and osteoarthritis outcome score (HOOS): 0 = extreme hip pain and impaired function, 100 = no hip pain or impaired function. bCollected at baseline only. cUniversity of California Los Angeles (UCLA) activity score: 1 = wholly inactive, 10 = regular participation in high impact sports.

Supplemental Information 4 Changes in squat depth between baseline and follow-up.

Individual participant changes (follow-up minus baseline) in maximum squat depth relative to starting height (change in vertical position of the midpoint of the two sacral markers, expressed as a percentage of limb length, %). The number of participants who decreased or increased squat depth at follow-up relative to baseline in the Personalised Hip Therapy (blue) and hip arthroscopy (red) groups were similar.

Supplemental Information 5 Baseline and follow-up participant joint angles and moments.

Time-normalised continuous joint angles and moments at baseline and 12-months follow-up, and treatment group allocations.

Supplemental Information 6 Baseline and follow-up participant spatiotemporal and peak-to-peak excursion data.

Spatiotemporal and peak-to-peak joint excursion data at baseline and 12-months follow-up, and treatment group allocations.

Supplemental Information 7 Baseline and follow-up participant characteristic data.

Participant characteristic data at baseline and 12-months follow-up, and treatment group allocations.

Supplemental Information 8 Trial protocol.

Source: Murphy, N.J., Eyles, J., Bennell, K.L. et al. Protocol for a multi-centre randomised controlled trial comparing arthroscopic hip surgery to physiotherapy-led care for femoroacetabular impingement (FAI): the Australian FASHIoN trial. BMC Musculoskelet Disord 18, 406 (2017). https://doi.org/10.1186/s12891-017-1767-y

We would like to acknowledge the participants and staff who contributed their time and efforts towards this study. We would like to thank Dr. Peter Smith from I-MED Radiology Network, VIC for support with radiology for this study. We would also like to thank the following physiotherapists who delivered the PHT intervention in Victoria: Tony Beecroft, Melbourne Sports Physiotherapy, Essendon, VIC; Andrew Dalwood, Physioworks Camberwell, Camberwell, VIC; Daniel Yee, West Active, Werribee, VIC; Laura Parlby, LifeCare, Croydon, VIC; Chris Snell, VIC; Amy Cusworth, LifeCare Croydon, VIC; Daniel Zwolak, Physica, Ringwood, VIC; Melissa Allen, VIC, Wade Byrnes, Waurn Ponds Physiotherapy Clinic, Waurn Ponds, VIC.

Additional Information and Declarations

Competing Interests

Author Contributions

Human Ethics

Data Availability

Clinical Trial Registration

David Lloyd has received research support from Arthrex and Orthopediatrics on an Australian Research Council Industrial Training and Transformation Centre grant, and from Orthocell on MTPConnect BioMedTech Horizons grant and Australian Research Council Industry Linkage grant. David J. Hunter has received consulting fees for scientific advisory roles from Pfizer, Lilly, Merck Serono, TLCBio, Kolon Tissuegene and Novartis. Nadine Foster is funded through an Australian National Health and Medical Research Council (NHMRC) Investigator Grant (ID: 2018182). No other potential Conflicts of Interest have been declared by any other authors.

Tamara M. Grant conceived and designed the experiments, analyzed the data, prepared figures and/or tables, authored or reviewed drafts of the article, and approved the final draft.

David J. Saxby conceived and designed the experiments, authored or reviewed drafts of the article, and approved the final draft.

Claudio Pizzolato conceived and designed the experiments, authored or reviewed drafts of the article, and approved the final draft.

Trevor Savage conceived and designed the experiments, analyzed the data, authored or reviewed drafts of the article, and approved the final draft.

Kim Bennell conceived and designed the experiments, authored or reviewed drafts of the article, trial/intervention conception and design, and approved the final draft.

Edward Dickenson conceived and designed the experiments, authored or reviewed drafts of the article, trial/intervention conception and design, and approved the final draft.

Jillian Eyles conceived and designed the experiments, authored or reviewed drafts of the article, trial/intervention conception and design, and approved the final draft.

Nadine Foster conceived and designed the experiments, authored or reviewed drafts of the article, trial/intervention conception and design, and approved the final draft.

Michelle Hall conceived and designed the experiments, performed the experiments, authored or reviewed drafts of the article, and approved the final draft.

David Hunter conceived and designed the experiments, authored or reviewed drafts of the article, trial/intervention conception and design, and approved the final draft.

David Lloyd conceived and designed the experiments, authored or reviewed drafts of the article, and approved the final draft.

Rob Molnar performed the experiments, authored or reviewed drafts of the article, and approved the final draft.

Nicholas Murphy conceived and designed the experiments, authored or reviewed drafts of the article, trial/intervention conception and design, and approved the final draft.

John O’Donnell conceived and designed the experiments, performed the experiments, authored or reviewed drafts of the article, trial/intervention conception and design, and approved the final draft.

Parminder Singh performed the experiments, authored or reviewed drafts of the article, and approved the final draft.

Libby Spiers performed the experiments, authored or reviewed drafts of the article, and approved the final draft.

Phong Tran performed the experiments, authored or reviewed drafts of the article, and approved the final draft.

Laura E. Diamond conceived and designed the experiments, prepared figures and/or tables, authored or reviewed drafts of the article, and approved the final draft.

The following information was supplied relating to ethical approvals (i.e., approving body and any reference numbers):

Ethical approval was granted by St Vincent’s Hospital Human Research Ethics Committee (HREC Reference Number: HREC/14/SVH/343).

The following information was supplied regarding data availability:

The baseline and 12-month follow-up data are available in the Supplemental Files.

The following information was supplied regarding Clinical Trial registration:

Australian Clinical Trials Registration Number: ACTRN12615001177549.

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
