# Peer review of "Squatting biomechanics following physiotherapist-led care or hip arthroscopy for femoroacetabular impingement syndrome: a secondary analysis from a randomised controlled trial"

_PeerJ, doi:10.7717/peerj.17567_

## Round 0.1 · original submission · Minor Revisions

Minor revisions are required, but please address all reviewer comments appropriately. I concur with reviewer 1 that some numerical values (change in angles) are needed to get a better understanding of the outcome of the trial. I am not an expert in statistics, but the comment from reviewer 2 that the study is underpowered, deserves attention. If a statistically significant is found, I believe that the study was sufficiently powered for that effect. On the other hand, if no statistical significance is found, it will be important to show that the power of the study (type 2 error rate) was high enough.

Reviewer 1 ·

Basic reporting

The manuscript is very well presented.

Experimental design

Within the journal's scope and research question well defined. Methods well presented and clearly to a high standard.

Validity of the findings

Data provided and conclusions well stated.

Additional comments

This manuscript examines biomechanical changes (angles and moments) during squatting after either surgical or non-surgical treatment for FAIS. The manuscript is well presented and the authors should be commended for their efforts. I have a few minor comments below, and a few general questions about the study, that hopefully will help the authors refine the manuscript for publication.
General questions/comments
1: I acknowledge that this perhaps was not within the authors control, but the physio treatment seems rather minor in the number of sessions (6-10 sessions over 12-24 weeks). This obviously can not be changed now, but this is worth mention in the discussion section in the context of biomechanical changes after a small number of physiotherapy sessions (perhaps alongside the comment about whether changes were due to the natural history of FAIS?). It seems emphasis could be placed on the fact that similar changes were observed in biomechanics between a surgical procedure that changes the shape of the hip and 6 sessions of physiotherapy!
2: Squat depth: it states in the methods that the participants descended until the end of self-selected range, but it is interesting to note that it appears quite similar between baseline and follow up for both groups (based on the numbers in Table 2 and sentence in the results). I suppose my first question was going to be around the non-standardisation of squat depth/height and how that might influence your findings, but given no differences, did many individuals squat with more or less depth at follow up (at the individual rather than group level)? Would you see different results if you had a standardised height/depth for both time points?

Other comments:
Abstract (methods): can you mention in the methods of the abstract what the squat depth was and whether it was consistent across time points?
Abstract (results): can you provide some numerical data in this section to help the reader? You mention significantly greater/reduced angles, but it would be nice to know if these changes are small 1-2 degrees or larger changes.

Introduction and Methods: very clear and well written – easy to follow so well done.

Results (Table 1): check the unit of measurement for BMI.
Results (lines 286 to 291): can you also provide mean differences for these comparisons?
Results (Table 2): sorry I may have missed it in the pdf I have, but can you tell the reader what positive/negative mean differences mean for both the within and between group comparisons?

Discussion (limitations): As noted above, is it worth mentioning here either the number of physiotherapist sessions and/or the squat depth in here?

Reviewer 2 ·

Basic reporting

Introduction:

Lines 77-78. The authors state that deep squatting is a complex multiplanar task. I do contend that the double leg squat is a complex multiplanar movement, since most of the joint kinematics are greatest in the sagittal plane across all lower extremity joints, pelvis and trunk. I agree that FAI is a 3D problem, however, a double leg squat predominantly evaluate sagittal plane motions. In lines 185 -188, the authors report how a heel wedge was used to "encourage deep hip flexion" by eliminating the need for ankle dorsiflexion. The fact that the authors chose this technique for the squat supports that there was considerable interest in the sagittal plane component of the movement. Although the hip moves in multiple planes during most tasks, I think it is an overstatement to say a double leg squat is multiple planar considering both feet are on the ground in a symmetrical fashion. Also for the double leg squat to be performed successfully the individuals needs to maintain double leg stability, while having adequate sagittal plane mobility to complete the task.

A single-leg squat task is a true multiplanar task, as this task inherently increases the demand for hip control in each plane during the tasks. Data on this task does exists n the FAIS population both before and after hip arthroscopy (See references below)

Malloy P, Wichman DM, Garcia F, Espinoza-Orías A, Chahla J, Nho SJ. Impaired lower extremity biomechanics, hip external rotation muscle weakness, and proximal femoral morphology predict impaired single-leg squat performance in people with FAI syndrome. The American Journal of Sports Medicine. 2021 Sep;49(11):2984-93.

Swindell H, Wichman DM, Guidetti M, Chahla J, Nho SJ, Malloy P. Association of Changes in Hip and Knee Kinematics During a Single-Leg Squat With Changes in Patient-Reported Outcomes at 6 Months and 1 Year After Hip Arthroscopy. The American Journal of Sports Medicine. 2023 Nov;51(13):3439-46.

I suggest the authors eliminate wording the states the double leg squat is a multiplanar task. Additionally the authors waveform data shows that as the hip is being flexed during squat descent, the joint is also abducting and externally rotating (Figure 4). Therefore, this double leg squat task does not simulate the impingement position of flexion, adduction, and internal rotation. Additionally, although not reported the magnitudes of the total hip joint excursion in the frontal and transverse planes, look to be less than 8 or so degrees, compared to almost 50-60 degrees in the sagittal plane, further supporting that a double leg squat is predominantly a sagittal plane, not multiplanar task.

Experimental design

Overall the experimental design is sound, however, this study is clearly underpowered as it was a secondary analysis from a clinical trial.

Validity of the findings

No comment

---

## Round 0.2 · Minor Revisions

Reviewer had additional comments which I would like you to address. Reviewer 1 would like to see their original comments 1 and 2 addressed in the manuscript, not just in the response to reviewer comments. In your response, please mention which change you made in the manuscript. Reviewer 1 also clarified their initial comment 3, which can hopefully be addressed better in a minor revision.

If I can see that you addressed the comments, I will decide to accept the manuscript and no additional review will be needed.

Reviewer 1 ·

Basic reporting

Clear.

Experimental design

Well defined.

Validity of the findings

Clear.

Additional comments

Thank you for the revisions.

For my first two comments in the original review, I would still encourage mention of both of these within the manuscript - the responses you provided are informative and interesting, and I believe should be provided to the future readers of the manuscript (1. number of sessions in each group mentioned in the discussion/limitations; and 2. the change in squat depth in the results/supplementary files).

Regarding comment 3 of my original review, you may have misread this or maybe I wasn't super clear, but I was suggesting that you should mention the squat depth in the abstract.

Reviewer 2 ·

Basic reporting

no comment

Experimental design

no comment

Validity of the findings

no comment

---

## Round 0.3 · accepted · Accept

Thank you for addressing the final comments of reviewer 1. I reviewed the response and the changes in the manuscript, and I am happy with the changes.

The manuscript is now ready for publication.